# Revealing a New Family of D-2-Hydroxyglutarate Dehydrogenases in *Escherichia coli* and *Pantoea ananatis* Encoded by *ydiJ*

**DOI:** 10.3390/microorganisms10091766

**Published:** 2022-08-31

**Authors:** Victor V. Samsonov, Anna A. Kuznetsova, Julia G. Rostova, Svetlana A. Samsonova, Mikhail K. Ziyatdinov, Michael Y. Kiriukhin

**Affiliations:** Ajinomoto-Genetika Research Institute, 117545 Moscow, Russia

**Keywords:** L-serine biosynthesis, D-3-phosphoglycerate dehydrogenase, D-2-hydroxyglutaric acid, D-2-hydroxyglutarate dehydrogenase

## Abstract

In *E. coli* and *P. ananatis*, L-serine biosynthesis is initiated by the action of D-3-phosphoglycerate dehydrogenase (SerA), which converts D-3-phosphoglycerate into 3-phosphohydroxypyruvate. SerA can concomitantly catalyze the production of D-2-hydroxyglutarate (D-2-HGA) from 2-ketoglutarate by oxidizing NADH to NAD^+^. Several bacterial D-2-hydroxyglutarate dehydrogenases (D2HGDHs) have recently been identified, which convert D-2-HGA back to 2-ketoglutarate. However, knowledge about the enzymes that can metabolize D-2-HGA is lacking in bacteria belonging to the *Enterobacteriaceae* family. We found that *ydiJ* encodes novel D2HGDHs in *P. ananatis* and *E. coli*, which were assigned as D2HGDH*Pa* and D2HGDH*Ec*, respectively. Inactivation of *ydiJ* in *P. ananatis* and *E. coli* led to the significant accumulation of D-2-HGA. Recombinant D2HGDH*Ec* and D2HGDH*Pa* were purified to homogeneity and characterized. D2HGDH*Ec* and D2HGDH*Pa* are homotetrameric with a subunit molecular mass of 110 kDa. The pH optimum was 7.5 for D2HGDH*Pa* and 8.0 for D2HGDH*Ec*. The Km for D-2-HGA was 208 μM for D2HGDHPa and 83 μM for D2HGDH*Ec*. The enzymes have strict substrate specificity towards D-2-HGA and displayed maximal activity at 45 °C. Their activity was completely inhibited by 0.5 mM Mn^2+^, Ni^2+^ or Co^2+^. The discovery of a novel family of D2HGDHs may provide fundamental information for the metabolic engineering of microbial chassis with desired properties.

## 1. Introduction

L-Serine biosynthesis is a substantial metabolic pathway in almost all living organisms, including *Escherichia coli* and *Pantoea ananatis*. L-Serine is a pivotal amino acid since it is used in protein synthesis and serves as a direct precursor for the biosynthesis of L-cysteine, L-methionine, L-tryptophan, and glycine. As the primary precursor of glycine, L-serine contributes a one-carbon unit (C1) that acts as a donor in methylation reactions via derivatives of tetrahydrofolate and S-adenosyl methionine. Thus, directly or indirectly, L-serine is a source of one-carbon units for the biosynthesis of various compounds, such as phosphatidylserine, sphingolipids, purines, and porphyrins [1,2,3,4,5,6]. It was also elucidated recently that L-serine synthesis through D-3-phosphoglycerate dehydrogenase (PHGDH) coordinates nucleotide levels in mammals by maintaining central carbon metabolism [7]. The phosphorylated glycolytic intermediate D-3-phosphoglycerate (3PG) is a bifurcation point of carbon flow for later steps in glycolysis toward pyruvate and L-serine biosynthesis. The pathway to L-serine biosynthesis is one of the main pathways in bacteria growing on sugars. For example, in *E. coli*, approximately 15% of the assimilated glucose carbon passes through L-serine before incorporation into biosynthetic products [8]. Three enzymes are responsible for bacterial de novo L-serine biosynthesis (Figure 1). D-3-Phosphoglycerate dehydrogenase (SerA, PHGDH) converts 3PG into phosphohydroxypyruvate (PHP), accompanied by the reduction of NAD^+^ to NADH. L-Phosphoserine aminotransferase (SerC, PSAT) converts PHP to L-phosphoserine (PS) with the concomitant deamination of glutamate to 2-ketoglutarate (2-KG). Finally, L-phosphoserine phosphatase (SerB, PSP) dephosphorylates PS, yielding L-serine [1,2,3,4,5,6,7,8,9].

PHGDH of *E. coli* (SerA), like many enzymes that catalyze the first key step in a biosynthetic pathway, is allosterically inhibited by the end product (L-serine) and by glycine, whose biosynthesis is linked to that of L-serine [9,10,11]. Dehydrogenation of 3PG, catalyzed by SerA, is thermodynamically unfavorable (ΔG° = +33.0 kJ·mol^−1^); the equilibrium of the reaction lies in the direction of D-3-phosphoglycerate [12]. As conventionally assumed, 3PG dehydrogenation is driven by coupling with reactions catalyzed by L-phosphoserine aminotransferase (SerC) (ΔG° = −11.5 kJ·mol^−1^) [12] and L-phosphoserine phosphatase (SerB) (ΔG° = −10.2 kJ·mol^−1^) [13] in L-serine biosynthesis; nevertheless, the calculated sum of ΔG° indicates that these reactions are not thermodynamically favorable [14]. According to the most plausible view, the reaction proceeds in the direction of L-serine synthesis because L-serine, the final product in that direction, is continually utilized in subsequent metabolic steps. This is proposed as a mechanism that conserves 3PG for later steps in glycolysis by using it only when L-serine is required [2,9].

In 1996, Zhao and Winkler [15] discovered that *E. coli* SerA could utilize 2-ketoglutarate (2-KG) as a substrate in the reverse direction in place of PHP, converting it to D-2-hydroxyglutarate (D-2-HGA) with the concomitant oxidation of NADH to NAD^+^ (Figure 1).

For a long time, the 2-ketoglutarate reductase activity of SerA was considered “promiscuous” and did not draw much attention after being identified. Just recently, the ability to use 2-KG as a substrate has also been reported for human PHGDH (hsPHGDH) [16,17], *Saccharomyces cerevisiae* (scPHGDH) [18], *Arabidopsis thaliana* (atPHGDH) [19], *Pseudomonas stutzeri* (psPHGDH) [14,20], *Pseudomonas aeruginosa* (paPHGDH) [14], and *Achromobacter denitrificans* (adPHGDH) [21]. Those findings led to the hypothetical proposal that D-2-hydroxyglutarate (D-2-HGA) production from 2-KG by PHGDH of *E. coli* (SerA) is necessary to convert the bound NADH to NAD^+^ in order to shift the reaction to proceed toward L-serine biosynthesis [22]. It is interesting that the ability to reduce 2-KG to D-2-HGA appears to be a common feature of Type II PHGDHs, while Type I or Type III PHGDHs examined so far do not share this feature [2,9,23,24].

In their recent innovative work, Zhang et al. [14] showed that in *Pseudomonas stutzeri* A1501, both 3PG dehydrogenation and 2-KG reduction are catalyzed by SerA, and SerA couples with the energetically favorable reaction of D-2-hydroxyglutaric acid (D-2-HGA) production from 2KG to overcome the thermodynamic barrier of 3PG dehydrogenation. They also identified a bacterial D-2-HGA dehydrogenase (D2HGDH), a flavin adenine dinucleotide (FAD)-dependent enzyme that subsequently converts D-2-HGA back to 2-KG. Electron transfer flavoprotein (ETF) and ETF-ubiquinone oxidoreductase (ETFQO) are also essential in D-2-HGA metabolism due to their capacity to transfer electrons from D2HGDH. Thus, it was uncovered that D-2-HGA-mediated coupling between SerA and D2HGDH drives bacterial L-serine biosynthesis [14].

As the key enzyme involved in D-2-HGA metabolism, D2HGDHs have been identified and characterized in bacteria (*Pseudomonas stutzeri* [14,20], *Pseudomonas aeruginosa* [14,25], *Achromobacter denitrificans* [21], and *Ralstonia solanacearum* [26]), yeast (*Saccharomyces cerevisiae* [18]), plants (*Arabidopsis thaliana* [27]), and humans [28,29,30].

Even though SerA of *E. coli* belongs to Type II PHGDHs and was reported to have the ability to produce D-2-HGA, there are neither D2HGDH nor ETF homologs in *E. coli*, suggesting the existence of other unknown enzymes in bacteria of the *Enterobacteriaceae* family involved in the possible D-2-HGA catabolism [14].

Due to that flaw, G. Grant proposed a self-sustaining cycle in *E. coli* L-serine biosynthesis that results in the conservation of NAD*^+^* and does not require D2HGDH [24]; nevertheless, this proposal does not explain the further fate of D-2-HGA that has to be produced via *E. coli* SerA during L-serine biosynthesis.

Thus, the goal of our work was to reveal possible mechanisms and enzymes, if any, that can metabolize D-2-HGA in *E. coli* and *P. ananatis* bacteria belonging to the *Enterobacteriaceae* family.

To accomplish this goal, we performed screening of the genomic *P. ananatis* library for genes that conferred fast growth on D-2-HGA. We detected D2HGDH activity in a crude extract of the clone containing the pSTV29 plasmid with *ydiJ*. Based on this observation, we proposed that the product of *ydiJ* can function as D2HGDH. Inactivation of this gene in both *E. coli* and *P. ananatis* led to a significant accumulation of D-2-HGA in culture media. These results indicate that YdiJ appears to be the only D2HGDH in *E. coli* and *P. ananatis*, providing the way to recuperate D-2-HGA back to 2-KG. The YdiJ genes of *E. coli* and *P. ananatis* were sequenced, cloned, and expressed in *Escherichia coli* as recombinant His-tagged proteins. We provide experimental evidence that *ydiJ* indeed encodes a novel family of bona fide D2HGDHs. The detailed enzymatic characterization of D2HGDH*Ec* and D2HGDH*Pa* adds a new and interesting member to the D2HGDH family.

## 2. Materials and Methods

### 2.1. Chemicals

D-2-Hydroxyglutarate (disodium salt), L-2-hydroxyglutarate (disodium salt), L-malate, D-malate, L-lactate, D-lactate, L-glycerate, D-glycerate, L-tartrate, D-tartrate, pyruvate, glutarate, Phenazine ethosulfate (PES), 2,6-Dichlorophenolindophenol (DCIP), isopropyl-β-D-1-thiogalactopyranoside (IPTG), TRIS, Bis-TRIS, and M9 minimal media were purchased from Sigma-Aldrich (San Luis, MO, USA). Yeast extract and tryptone were obtained from BD Biosciences (San Jose, CA, USA). Other chemicals were of analytical reagent grade.

### 2.2. Analytical Method

The analysis of organic acids, including D-2-HGA, was carried out by HPLC (Shimadzu system IE-HPLC-ECD) using a CDD-10A conductivity detector. Generally, a Phenomenex Rezex ROA-Organic Acid H+, 8% column was used, and the mobile phase (5 mM p-toluenesulfonic acid monohydrate, 100 mM EDTA-2Na, and 20 mM BIS-TRIS, pH 6.0) was pumped at a flow rate of 0.8 mL/min and a temperature of 40 °C.

### 2.3. Bacterial Strains, Plasmids, and Growth Conditions

The strains and plasmids used in this study are listed in Table 1. All strains were grown at 34 °C in Luria–Bertani (LB) medium or M9 minimal medium [31] supplemented with 25 μg/mL—kanamycin, 25 μg/mL—chloramphenicol, and 20 μg/mL—tetracycline if required. M9 minimal medium with the addition of 5 g/L D-2-Hydroxyglutarate (disodium salt) (D-2-HGA) was used for the selection of clones containing gene bank plasmids with targeted genes growing on D-2-HGA. The test tube fermentation medium for the determination of D-2-HGA accumulation was composed of M9 supplemented with Glucose (40 g/L), CaCO_3_ (20 g/L), Tryptone (0.6 g/L), Yeast extract (0.3 g/L), and chloramphenicol 25 μg/mL, and the medium for the determination of the growth rate was composed of M9 supplemented with D-2-HGA 5 g/L.

### 2.4. DNA Manipulation and Plasmid Construction

Plasmid DNA was isolated using the Cleanup Standard kit (Evrogen, Moscow, Russia). Chromosomal DNAs from *E. coli* and *P. ananatis* were isolated using the Wizard Genomic DNA Purification Kit (Promega, Madison, WI, USA). Restriction endonucleases, Klenow Fragment, T4 DNA ligase, and Taq DNA polymerases were purchased from Thermo Fisher Scientific Inc. (Waltham, MA, USA) or New England Biolabs (Ipswich, MA, USA) and used according to the manufacturer’s instructions. *E. coli* and *P. ananatis* cells were transformed by a standard electroporation procedure using the MicroPulser Electroporator (BioRad, Hercules, CA, USA).

The *ydiJ* genes from *E. coli* and *P. ananatis* were PCR-amplified by KAPA HiFi DNA Polymerase (Roche, Basel, Switzerland) from genomic DNA of *E. coli* MG1655 and *P. ananatis* SC17 (AJ13355) using the primers listed in Appendix A. Inactivation or enhancement of *ydiJ* expression in *P. ananatis* or *E. coli* was performed using the λ-Red recombination system [33]. The plasmid pRSFRedTER [GenBank: FJ347161], which carries an IPTG-inducible λ-Red gene, was used to allow Red recombination. The plasmid pMW-λattL-Km-R-λattR (Km) was used as a template to provide PCR-generated gene disruption cassettes, and pMW-λattR-Km-R-λattL-P_nlp8_ (Km^r^) was used to enhance gene expression [33]. The primers used to generate these cassettes are listed in Appendix A.

For overexpression of genes in chromosomes, randomized in -10 and -35 regions, a derivative of the constitutive promoter of the *E. coli nlpD* gene was used in the present work [37]. The antibiotic resistance marker Km was introduced upstream of the promoter. The direction of Km transcription was opposite to the direction of transcription from the P_nlp8_ promoter. P_nlp8_ is a stronger promoter than P_nlpD_ [37]. A derivative of the SerA (3-Phosphoglycerate dehydrogenase) enzyme from *E. coli* (SerA348) with N348A replacement was used. The SerA348 enzyme is feedback-resistant to L-serine [38]. To construct the *serA*348 expression plasmid, a low-copy-number plasmid, pMIV-5JS, was used as a backbone. The PCR fragment containing the P_nlp8_*serA*348 expression cassette was digested using *PaeI* and *XbaI* (these sites were designed based on the 5′ and 3′ ends of the primers) and cloned into the corresponding sites in pMIV-5JS.

To obtain YdiJ with an N-terminal His-tag, the pairs of primers *ydiJ*_EcF1-*ydiJ*_EcR1 and *ydiJ*_PaF1-*ydiJ*_PaR1 were used for PCR amplification of *ydiJ* from chromosomes of *E. coli* and *P. ananatis*, resulting in PCR fragments *ydiJ*_Ec1 and *ydiJ*_Pa1, respectively. For the C-terminal His-tag in YdiJ, the pairs of primers *ydiJ*_EcF2-*ydiJ*_EcR2 and *ydiJ*_PaF2-*ydiJ*_PaR2 were used and resulted in *ydiJ*_Ec2 and *ydiJ*_Pa2 fragments. After digestion with *Bsp*119I, the *ydiJ*_Ec1 fragment was blunted by the Klenow Fragment, digested with *XhoI*, and cloned into *NdeI* (blunted) and *XhoI* sites of pET28b(+); the resulting plasmid was pET-*ydiJ*-Ec1. Plasmids pET-*ydiJ*-Ec2 and pET-ydiJ-Pa1 were obtained by cloning fragments with *ydiJ* genes from *E. coli* and *P. ananatis* into *NcoI*, *BamHI* and *NdeI*, *XhoI* sites of pET28b(+), respectively. The plasmid pET-*ydiJ*-Pa2 was obtained by cloning the *ydiJ*_Pa2 fragment digested with *PscI* and *XhoI* into *NcoI*, *XhoI* sites of pET28b(+). The integrity of the nucleotide sequence of all newly constructed plasmids was confirmed by DNA sequencing. The obtained plasmids were transformed into BL21 (DE3) cells for protein expression.

### 2.5. Screening for Genes Involved in D-2-HGA Utilization

Genomic DNA from wild-type *P. ananatis* strain SC17 was partially digested with *Sau3AI*, and fragments measuring approximately 7 kb were cloned into the *BamHI* site of the multicopy vector pSTV29 (Takara Bio, Shiga, Japan). The genomic library obtained was introduced into SC17 by electroporation. Transformed cells were then challenged on M9 plates containing 25 μg/mL chloramphenicol and 5 g/L D-2-HGA for 2–4 days. Fast-growing colonies were isolated, and candidate genes in the plasmids that conferred faster growth on D-2-HGA were analyzed by sequencing the plasmid DNA.

### 2.6. DNA Analysis

Sequencing of both strands was performed using the dideoxynucleotide chain-termination method using the oligonucleotide primers of the vector pSTV29. Primers were synthesized according to presequenced regions and used for progressive sequencing. Sequence comparisons were made with EMBL-EBI services’ “Clustal Omega” and NCBI services.

### 2.7. Enzyme Overexpression and Purification

The *E. coli* BL21 (DE3) strain (picked from a single colony) harboring pET-*ydiJ*-Ec1, pET-*ydiJ*-Pa1 (N-terminal 6His-tags), pET-*ydiJ*-Ec2, and pET-*ydiJ*-Pa2 (C-terminal 6His-tags) was propagated overnight in Luria–Bertani (LB) medium with 50 μg/mL kanamycin at 30 °C with shaking on a rotatory plate at 240 rpm. All of the overnight growth cultures were used to inoculate 1 L (4 × 250 mL) of fresh LB to a final OD600nm of 0.1 with the same antibiotic to grow until the cell density reached an OD600nm of 0.5–0.6. IPTG was added to the culture at a final concentration of 1 mM, and the growth continued at 30 °C for about 105 min. Cells were harvested by centrifugation at 10,000× *g* for 10 min at 4 °C, washed with buffer A (20 mM Tris-HCl, 20 mM Imidazole-HCl, and 0.5 M NaCl, pH 8.0), and resuspended in 30 mL of the same buffer. The cells were disrupted by French press (Thermo Fisher) at 4 °C until clear lysate was obtained. The cell debris was then removed by centrifugation at 12,000× *g* for 20 min at 4 °C. The recombinant YdiJ with a 6x His-tag on its N- or C-terminus was purified using a 5 mL HisTrap HP column (GE Healthcare Life Sciences) equilibrated with buffer A. Unbound protein was washed away with buffer A. The protein fractions were eluted with an imidazole gradient of 0 to 100% buffer B (20 mM Tris-HCl, 500 mM Imidazole-HCl, and 0.5 M NaCl, pH 8.0) and 500 mM elution buffer. The active fraction with the recombinant D2HGDH was pooled and concentrated, and the buffer was changed to 25 mM Tris-HCl, pH 7.5, using a Vivaspin 20 centrifugal concentrator. The purity of the recombinant enzymes was confirmed by 12% SDS-PAGE and non-denaturing 4–20% gradient PAGE.

### 2.8. Measurement of Enzyme Activity

D-2-Hydroxyglutarate dehydrogenase (D2HGDH) activity was routinely measured by following the reduction of DCIP spectrophotometrically at 600 nm [14]. Reaction mixtures were incubated at 25 °C and contained 50 mM Tris-HCl buffer (pH 7.5), 200 μM phenazine ethosulfate (PES), 100 μM DCIP, 1 mM D-2-HGA (disodium salt), and cell extract (or pure enzyme) in a total volume of 1.0 mL. Phenazine ethosulfate (PES) was used instead of phenazine methosulfate (PMS) due to its higher stability, especially at increased pH and ionic strength [39]. After the determination of the pH optimum, D2HGDH*Pa* was measured in 50 mM Tris-HCl buffer (pH 7.5), while D2HGDH*Ec* was measured in 50 mM Tris-HCl buffer (pH 8.0). The reduction of DCIP was monitored at 600 nm with a thermostated Shimadzu-1800 UV-Vis spectrophotometer (Shimadzu Corp, Kyoto, Japan), converting the absorbance to concentration using a molar extinction coefficient of 22 mM^−1^cm^−1^. One unit (U) of activity was defined as 1 μmol of DCIP reduced per minute. Protein concentrations were determined using the Bio-Rad protein assay kit (Bio-Rad, Hercules, CA, USA) with bovine serum albumin as a standard. All measured values indicate the means of at least three independent experiments.

### 2.9. Characterization of the Recombinant D2HGDH (YdiJ)

The native molecular mass of the recombinant D2HGDH*Ec* and D2HGDH*Pa* were estimated by non-denaturing 4–20% gradient gel electrophoresis. The protein standards used for the calibration of the gel were albumin (66 kDa), lactate dehydrogenase (140 kDa), catalase (250 kDa), ferritin (440 kDa), and thyroglobulin (669 kDa) (GE Healthcare Life Sciences).

The effects of pH and temperature on recombinant D2HGDH*Ec* and D2HGDH*Pa* activity were determined in the standard reaction mixture. To obtain the pH profile, the enzyme was assayed in 100 mM buffer (Bis-Tris–HCl, pH 5.5–7.0) and (Tris–HCl, pH 7.5–9.0).

The temperature optimum was determined at various temperatures up to 60 °C. The temperature influence on protein stability was investigated by means of pure enzyme (0.03 mg/mL) incubation in 50 mM Tris-HCl, pH 7.5, at different temperatures for 10 min, after which aliquots were immediately cooled on ice, and the residual activity was assayed. The kinetic parameters for the recombinant D2HGDH*Ec* and D2HGDH*Pa* were determined by measuring their activity at various D-2-HGA concentrations at saturating concentrations of another substrate. The apparent kinetic parameters were calculated by a double-reciprocal Lineweaver–Burk plot.

The effects of different metal ions (0.5 mM MnCl_2_, 0.5 mM MgCl_2_, 0.5 mM CoCl_2_, 0.5 mM NiCl_2_, and 0.5 mM ZnSO_4_) or other substrates on the activity of D2HGDH*Ec* or D2HGDH*Pa* were determined using the standard assay protocol at the pH optimum.

### 2.10. Polyacrylamide Electrophoresis

Analysis by 12% SDS-PAGE was carried out as described elsewhere (https://www.bio-rad.com/webroot/web/pdf/lsr/literature/Bulletin_6201.pdf, accessed on 1 August 2022). Non-denaturing gradient PAGE (4–20%, Bio-Rad Mini-Protean^®^ GTG™ gel, #456-1093) was carried out in standard Tris-Glycine buffer without SDS and reducing agents. Gels were stained with Coomassie brilliant blue R-250 staining solution (Bio-Rad, Hercules, CA, USA).

### 2.11. Structure-Based Protein Sequence Alignment

Structure-based amino acid sequence alignment was conducted with the CLUSTALX program (ftp://ftp.ebi.ac.uk/pub/software/clustalw2, accessed on 1 June 2022) and ESPRIPT 3.0 web tool (http://espript.ibcp.fr/ESPript/ESPript/, accessed on 1 June 2022) [40,41].

### 2.12. Phylogenetic Analysis

The phylogenetic tree of YdiJs from *E. coli* and *P. ananatis* was constructed by means of the NCBI BLASTP service to check its distribution among the *Proteobacteria* phylum.

## 3. Results and Discussion

### 3.1. Screening the Genomic P. ananatis Library for Genes That Conferred Fast Growth on D-2-HGA

To elucidate the ability of *P. ananatis* to utilize D-2-HGA as a sole carbon source, we plated a wild-type strain of *P. ananatis* SC17 (AJ13355) on M9 minimal agar supplemented with 5 g/L D-2-HGA. Tiny colonies appeared after 7–8 days of incubation at 34 °C, so we can conclude that D-2-HGA can support the growth of *P. ananatis*, although it barely serves as a good substrate. To identify *P. ananatis* genes encoding possible enzymes that take part in D-2-HGA utilization, we screened a multicopy genomic library prepared from wild-type *P. ananatis* genomic DNA in a pSTV29 vector. *P. ananatis* strain SC17 (AJ13355) was electroporated with the genome library, and transformed cells were cultured on M9 minimal agar supplemented with 5 g/L D-2-HGA and 25 μg/mL chloramphenicol. Fast-growing colonies were selected after 2–4 days, and the genes responsible for the fast-growing phenotype were identified by sequencing the plasmid DNAs. Of thirty analyzed clones, twenty-nine sequenced fragments had obscure ORF or ORF with predicted membrane proteins that are under our investigation at the moment as possible importers of D-2-HGA. One of the sequenced ORF fragments contained the *ydiJ* gene (NCBI Reference Sequence: WP_019105315.1) with high similarity to *E. coli ydiJ* (NCBI Reference Sequence: NP_416202.1), encoding a polypeptide of 1018 amino acids with identities of 79% (Figure 2). The predicted pI values are 7.80 and 7.11 for YdiJ from *P. ananatis* and *E. coli*, respectively. While D2HGDH activity was not detected in wild-type *P. ananatis* even when it grew on D-2-HGA, we were able to detect D2HGDH activity at 0.87 × 10^−3^ U/mg protein in a crude extract of a clone containing the pSTV29 plasmid with *ydiJ*. Based on this observation, we proposed that the product of *ydiJ* could function as D2HGDH, and we assigned YdiJs of *E. coli* and *P. ananatis* as D2HGDH*Ec* and D2HGDH*Pa*, respectively.

In *E. coli* and *P. ananatis*, the product of *ydiJ* was described as a putative cytosolic FAD-linked oxidoreductase of unknown function. It is interesting that YdiJ was recently predicted to be a metalloprotein in *E. coli* and isolated; the function of the protein was not reported. As isolated, purified YdiJ contains FAD as well as a 4Fe-4S cluster [42]. A comparison of the domain structure of YdiJ and known D2HGDH from *P. stutzeri* A1501 reveals that YdiJ is about two times longer and contains an additional GlpC superfamily domain, which is the membrane-associated subunit of the heterotrimeric glycerol-3-phosphate dehydrogenase complex (Figure 3). Multiple sequence alignment of the first 542 residues (GlcD superfamily, FAD oxidoreductase domain) of YdiJs (D2HGDHs) of *P. ananatis* (*Pa*) and *E. coli* (*Ec*) with the whole sequence of D2HGDH of *P. stutzeri* (*Ps*) reveals their very weak homology, with identities of about 23–25% (Figure 4).

### 3.2. Determination of Physiological Function of YdiJ In Vivo

To determine the in vivo function of YdiJ in *E. coli* and *P. ananatis*, we inactivated the corresponding genes, resulting in *E. coli* strain MG1655Δ*ydiJ*::Km and *P. ananatis* strain SC17(0)Δ*ydiJ*::Km. They were evaluated in conditions of test tube fermentation, as described in Materials and Methods. Compared to the parental strains, test tube fermentation of these strains revealed that inactivation of *ydiJ* led to the accumulation of D-2-HGA to 4.2 g L^−1^ for MG1655Δ*ydiJ*::Km and 2.7 g L^−1^ for SC17(0)Δ*ydiJ*::Km (Figure 5A). Thus, we demonstrated that disruption of *ydiJ* in both *E. coli* and *P. ananatis* impairs D-2-HGA utilization, which is expected to be produced via 2KG reductase activity of SerA (PHGDH).

To determine the influence of the inactivation of *ydiJ* or its enhanced expression on D-2-HGA accumulation in strains with high-level expression of *serA*, we constructed the plasmid pMIV-P_nlp8_*serA*348 with high-level expression of a feedback-resistant variant of *E. coli* SerA (SerA348) [38] by introducing a P_nlp8_*serA*348 cassette in pMIV-5JS to enhance D-2-HG synthesis in cells. The plasmid pMIV-P_nlp8_*serA*348 was transformed into *P. ananatis* SC17(0), SC17(0)Δ*ydiJ*::Km, SC17(0)P_nlp8_*ydiJ*_Pa_, *E. coli* MG1655, MG1655Δ*ydiJ*::Km, and MG1655P_nlp8_*ydiJ_Ec_*, and the resulting strains were evaluated in test tube fermentation. D-2-HGA accumulation was determined after fermentation for all tested strains and is shown in Figure 5B. Inactivation of *ydiJ* led to an enormous accumulation of D-2-HGA to 25 g L^−1^ for MG1655Δ*ydiJ*::Km and 12 g L^−1^ for SC17(0)Δ*ydiJ*::Km.

### 3.3. Recombinant Enzyme Purification and Characterization

Although four plasmids were constructed for the overexpression of *ydiJ* from *E. coli* and *P. ananatis*, many attempts to isolate the proteins failed because they tended to form inactive inclusion bodies without the detection of corresponding D2HGDH activity in the soluble fraction and the corresponding protein band on SDS-PAGE. We performed a number of approaches to improve the expression of recombinant YdiJ in soluble form, which include changing the *E. coli* host, cultivation conditions, such as expression temperature, medium composition, the timing of induction, and inducer concentration. Finally, we found that decreasing the induction time to 90–100 min and quickly starting the purification procedure led to the most reliable results, although even under these conditions, most of the recombinant YdiJ from both *E. coli* and *P. ananatis* precipitated into inclusion bodies. The *E. coli* BL21 (DE3)-harboring plasmid pET-*ydiJ*_Ec1 (N-terminus 6xHis-tagged D2HGDHEc, Table 1) or pET-*ydiJ*_Pa2 (C-terminus 6xHis-tagged D2HGDHPa, Table 1) did not produce a detectable protein band in the soluble fraction corresponding to the size of the fusion protein under various experimental conditions.

Recombinant C-terminus 6xHis-tagged D2HGDHEc (*E. coli*, *ydiJ*_Ec2, Table 1) and D2HGDHPa N-terminus 6xHis-tagged D2HGDHPa (*P. ananatis*, *ydiJ*_Pa1, Table 1) were purified to homogeneity. The molecular mass of both enzymes was determined to be approximately 110.0 kDa by SDS-PAGE, which compares well to the predicted value (113 kDa) (Figure 6A). The oligomeric status of D2HGDH*Ec* and D2HGDH*Pa* was confirmed by non-denaturing 4–20% gradient PAGE, which showed one protein band for both D2HGDH*Ec* and D2HGDH*Pa* with a molecular mass of approximately 440 kDa, suggesting that the native enzyme forms a homotetramer in solution (Figure 6B). Unfortunately, gel filtration chromatography on a HiLoadTM 10/300 Superdex 200 column (GE Healthcare) resulted in a diffuse non-symmetrical peak, possibly due to protein aggregation in standard elution conditions.

The final specific activity of the purified D2HGDH*Ec* was 1.12 U mg^−1^ and 1.01 U mg^−1^ for D2HGDH*Pa*. The enzymes showed Michaelis–Menten kinetics. The apparent K_m_ value for D-2-HGA was 83 μM for D2HGDH*Ec* and 208 μM for D2HGDH*Pa* (Appendix A). All steady-state kinetic parameters of D2HGDH*Ec* and D2HGDH*Pa* are shown in Table 2.

The K_m_ value of D2HGDH*Ec* for D-2-HGA (83 μM) is a little bit higher than that determined for D2HGDHs from the bacteria *P. aeruginosa* (60 μM) [25] and *A. denitrificans* (32 μM) [21] but lower than that demonstrated for *P. stutzeri* (170 μM) and *R. solanacearum* (433 μM) [14,20,26]. The K_m_ value of D2HGDH*Pa* for D-2-HGA (208 μM) is basically within the range of characterized D2HGDHs from *P. stutzeri*, *A. thaliana*, *S. cerevisiae*, *R. solanacearum*, and Homo sapiens, as demonstrated in Table 2.

The catalytic efficiency of D2HGDH*Ec* (170 s^−1^mM^−1^) is very close to those in *P. aeruginosa* (186 s^−1^mM^−1^) and *A. denitrificans* (230 s^−1^mM^−1^) and higher than those of all other known D2HGDHs. In contrast, the catalytic efficiency of D2HGDH*Pa* (50 s^−1^mM^−1^) is much lower and comparable to the efficiency of characterized D2HGDHs from *P. stutzeri*, *S. cerevisiae*, *R. solanacearum*, and Homo sapiens.

### 3.4. Effects of pH and Temperature

The effects of pH and temperature on the activity of D2HGDH*Ec* and D2HGDH*Pa* were determined using the standard assay protocol. The results showed that the optimum pH was 7.5 for D2HGDH*Pa* and 8.0 for D2HGDH*Ec* (Figure 7A), which are in the range of all known D2HGDHs listed in Table 2. The temperature optimum was essentially the same (45 °C) for both enzymes (Figure 7B). This value is the same as for D2HGDHs from *R. solanacearum* and much lower than those for D2HGDH from *P. stutzeri* (70 °C) [14,26]. Heat-inactivation studies revealed that D2HGDH*Pa* and D2HGDH*Ec* have the same thermostability range (20–25 °C) and showed instability during incubation at temperatures higher than 25 °C. D2HGDH*Pa* and D2HGDH*Ec* activity decreased almost linearly in the temperature range from 30 °C to 60 °C (Figure 7C). This range is very close to that of D2HGDH in *P. stutzeri*, which also lost activity at temperatures higher than 37 °C [14].

### 3.5. Effects of Metal Ions on D2HGDH Activity

The effects of different cations (0.5 mM MnCl_2_, 0.5 mM MgCl_2_, 0.5 mM CoCl_2_, 0.5 mM ZnSO_4_, and 0.5 mM NiCl_2_) on enzyme activity were studied, and the results indicate that D2HGDH*Pa* and D2HGDH*Ec* activity was completely inhibited by the addition of 0.5 mM Mn^2+^, Ni^2+^, or Co^2+^ and partially inhibited by 0.5 mM Zn^2+^ (Table 3). It was shown that for D2HGDH from *P. stutzeri*, Zn^2+^ positively influenced activity at a concentration of 10 μM but inhibited D2HGDH activity at a concentration of 10 mM [20]. D2HGDH from *S. cerevisiae* (Dld2) is stimulated by 5 μM Zn^2+^, whereas Co^2+^, Mn^2+^, Mg^2+^, and Ca^2+^ did not affect its activity at this concentration. For Dld3, Zn^2+^ and Co^2+^ stimulated D2HGDH activity to a similar extent at the low and high metal concentrations tested, whereas Mn^2+^, Mg^2+^, and Ca^2+^ did not significantly affect its activity [18]. Rat liver D2HGDH is stimulated by 100 μM Zn^2+^, Co^2+^, and Mn^2+^, but not by Mg^2+^ or Ca^2+^ [35].

### 3.6. Substrate Specificity

Substrate specificity screening revealed that D2HGDH*Ec* and D2HGDH*Pa* have strict substrate specificity and only exhibited distinct activity towards D-2-HGA, but there was no detectable activity on L-2-hydroxyglutarate, L-malate, D-malate, L-lactate, D-lactate, L-tartrate, D-tartrate, L-glycerate, D-glycerate, glutarate, or pyruvate at final concentrations of 5–10 mM. This result is similar to those obtained for bacterial D2HGDHs from *A. denitrificans* [21] and *R. solanacearum* [26] and for plant D2HGDH from *A. thaliana* [27], which also demonstrated a strict preference for D-2-HGA. Conversely, D2HGDHs from bacteria *P. stutzeri* [14,20] and *P. aeruginosa* [25] demonstrated high detectable D-malate oxidizing activity, comparable to that of D-2-HGA. Moreover, it was proposed that D2HGDH evolved as an enzyme for both D-malate and D-2-HGA dehydrogenation in *P. stutzeri* A1501, and the enzyme participates in both the core metabolic pathway for L-serine biosynthesis and utilization of extracellular D-malate [20]. D2HGDH from *S. cerevisiae* (Dld2) displayed promiscuous activity toward D-malate, D-lactate, and even L-2-hydroxyglutarate. In addition to D-2-HGA and D-malate, Dld3 showed substantial activity with D-lactate and lower activity with D-glycerate; however, it displays a higher stereoselectivity for D-2-hydroxyacids over L-2-hydroxyacids than Dld2 [18]. Human D2HGDH exhibits high specific activities towards both D-2-HGA (2.02 ± 0.04 μmol/min/mg) and D-malate (2.52 ± 0.05 μmol/min/mg) and weak activity towards D-lactate (0.16 ± 0.01 μmol/min/mg) and L-2-hydroxyglutarate (0.06 ± 0.01 μmol/min/mg) [28,29]. D2HGDH isolated from rat liver also oxidized D-lactate, D-malate, and *meso*tartrate [35].

### 3.7. Phylogenetic Analysis of YdiJ-Like Enzymes

Analysis of YdiJ distribution confirmed that it could be D2HGDH for a large group in the *Proteobacteria* phylum, namely, for the class gamma *Proteobacteria*, with maximal identity to the *Enterobacteriaceae* family (Figure 8). Of 1000 analyzed species of the *Enterobacteriaceae* family, all had enzymes with a identity of more than 49% to D2HGDH*Pa* and D2HGDH*Ec*, and all had Type II SerA, which was able to generate D-2-HGA from 2KG.

D2HGDH*Ps* is also an enzyme from the *Proteobacteria* phylum. From 1000 analyzed species of the *Proteobacteria* phylum, the coexistence of D2HGDH*Ps*-like enzymes and Type II SerA was about 30%, and the coexistence of D2HGDH*Pa*-like or D2HGDH*Ec*-like enzymes and Type II SerA in this phylum was more than 65% (Appendix A). There are probably only two types of D2HGDH that exist in the *Proteobacteria* phylum. Thus, D2HGDH*Pa*-like or D2HGDH*Ec*-like enzymes are predominant in the *Proteobacteria* phylum.

## 4. Conclusions

In the present study, we report the identification and biochemical characterization of a novel family of D2HGDHs encoded by *ydiJ* in *P. ananatis* and *E. coli*. D-2-HGA has been detected in various organisms, including Homo sapiens, *Arabidopsis thaliana*, and *Saccharomyces cerevisiae*, and extensively studied after being identified as an “oncometabolite” in humans [14,16,45]. Although D-2-HGA has also been found at low detectable levels in bacteria such as *Pseudomonas* [46] and *E. coli* [47], it did not draw much attention because it was not considered a core metabolite. Recently, Zhang et al. [14] showed that D-2-HGA is a “normal” metabolite that is simultaneously produced by SerA (Type II PHGDH) and catabolized by D2HGDH without accumulation in bacterial metabolism in *P. stutzeri*. Moreover, they concluded that coupling between D-3-phosphoglycerate dehydrogenase and D2HGDH drives bacterial L-serine synthesis. In our study, we also demonstrated that D-2-HGA is a normal metabolite in *P. ananatis* and *E. coli* produced during L-serine synthesis by SerA and is subsequently converted back to 2KG via D2HGDH encoded by *ydiJ*. The physiological molecule that functions as the primary electron acceptor during D-2-HGA oxidation by YdiJ in *P. ananatis* and *E. coli* is unknown and requires further investigation (Figure 9). The discovery of a novel D2HGDH encoded by *ydiJ* adds a new and interesting member to the D2HGDH family and may provide fundamental information for metabolic engineering of microbial chassis with desired properties. Previously, we demonstrated the great impact of the overexpression of *ydiJ* on the production of L-cysteine or L-methionine by a *P. ananatis*-based microbial platform [48].

## Figures and Tables

**Figure 1 microorganisms-10-01766-f001:**
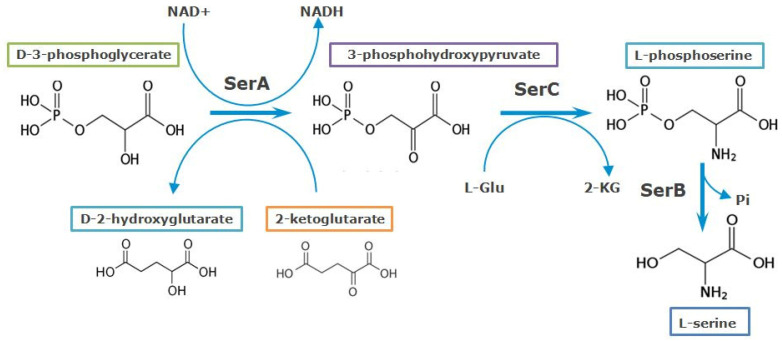
The pathway of L-serine biosynthesis in *E. coli*. 2-Ketoglutarate reductase activity of SerA is represented; P_i_—inorganic phosphate.

**Figure 2 microorganisms-10-01766-f002:**
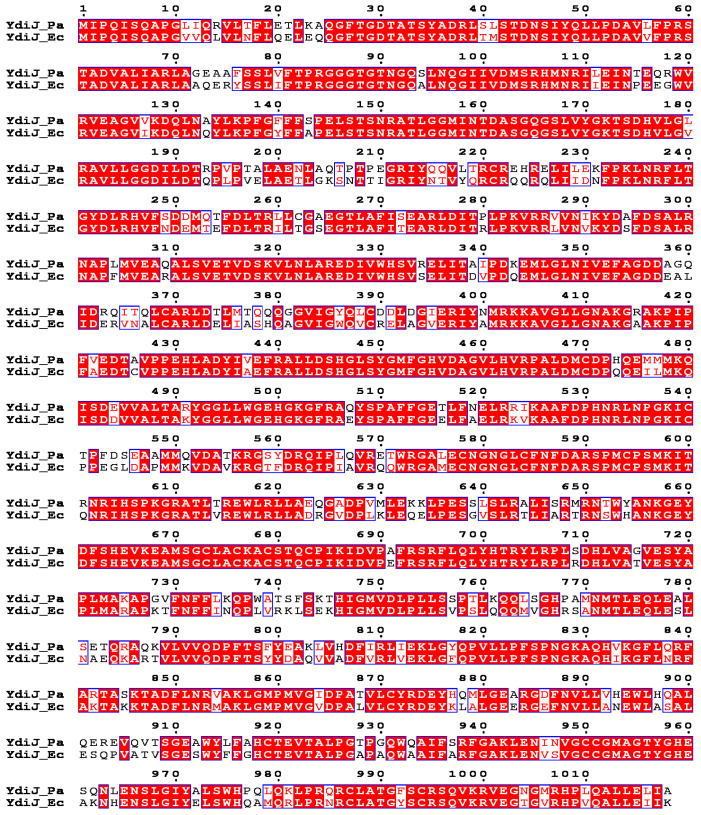
Sequence alignment of the YdiJs (D2HGDHs) of *P. ananatis* (*Pa*) and *E. coli* (*Ec*). The conserved amino acids are highlighted in shaded red boxes; conserved replacements are highlighted as red letters in blue boxes. The alignment was drawn with ESPRIPT 3.0.

**Figure 3 microorganisms-10-01766-f003:**
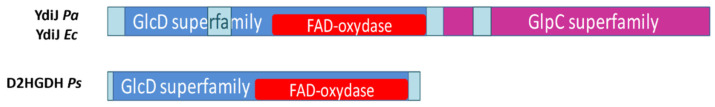
Domain structure of YdiJ of *P. ananatis* (*Pa*) and *E. coli* (*Ec*) and D2HGDH of *P. stutzeri* A1501 (*Ps*).

**Figure 4 microorganisms-10-01766-f004:**
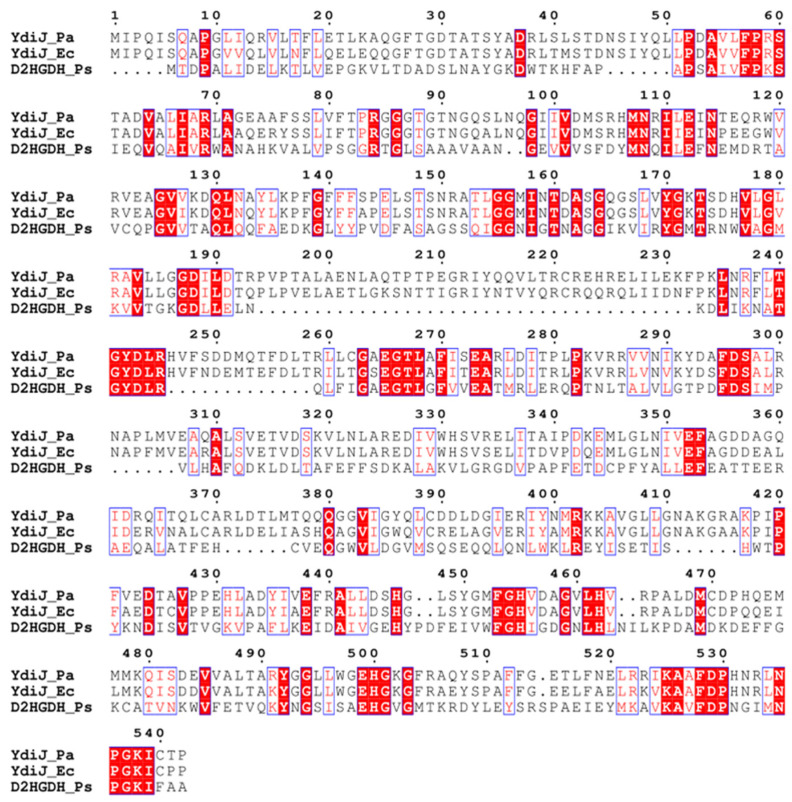
Multiple sequence alignment of the first 542 residues (GlcD superfamily, FAD oxidoreductase domain) of YdiJs (D2HGDHs) of *P. ananatis* (*Pa*) and *E. coli* (*Ec*) with the whole sequence of D2HGDH of P*. stutzeri* (*Ps*). The alignment was drawn with ESPRIPT 3.0.

**Figure 5 microorganisms-10-01766-f005:**
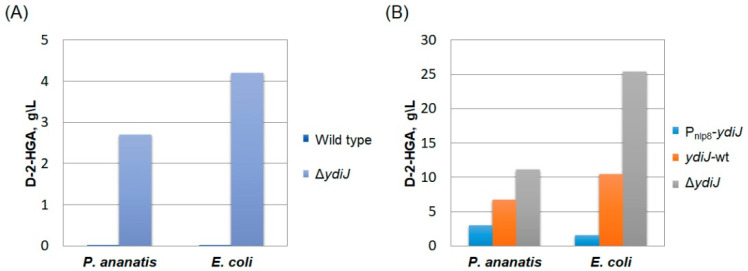
(**A**) D-2-HGA accumulation by *E. coli* MG1655Δ*ydiJ*::Km and *P. ananatis* SC17(0)Δ*ydiJ*::Km. (**B**) D-2-HGA accumulation by *P. ananatis* SC17(0)/pMIV-P_nlp8_-*serA*348 and *E. coli* MG1655/pMIV-P_nlp8_-*serA*348 with various allelic stages of the *ydiJ* gene: enhanced expression of the *ydiJ* gene (P_nlp8_-*ydiJ*), expression of native *ydiJ* (*ydiJ*-wt) and inactivation of *ydiJ* (Δ*ydiJ*).

**Figure 6 microorganisms-10-01766-f006:**
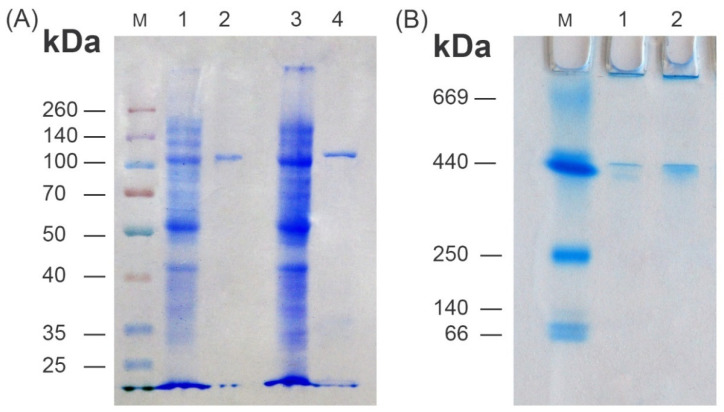
SDS-PAGE (**A**) and non-denaturing 4–20% gradient PAGE (**B**). (**A**) M—molecular weight marker (Spectra™ Multicolor Broad Range Protein Ladder, Thermo Scientific™, #26634); 1—crude extract of recombinant D2HGDH*Pa*; 2—purified D2HGDH*Pa*; 3—crude extract of recombinant D2HGDH*Ec*; 4—purified D2HGDH*Ec*; (**B**) M—molecular weight marker (GE Healthcare HMW Calibration Kit proteins, # 17-0445-01); 1—recombinant D2HGDH*Pa*; 2—recombinant D2HGDH*Ec*.

**Figure 7 microorganisms-10-01766-f007:**
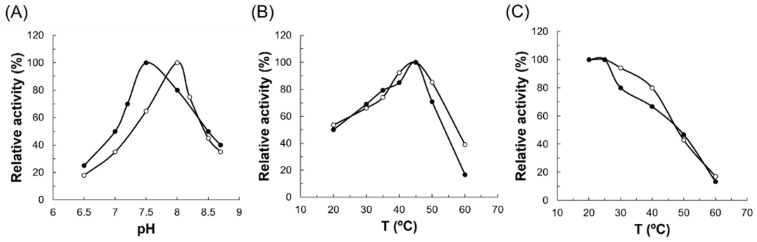
(**A**) Effects of pH on the activity of recombinant D2HGDH*Ec* and D2HGDH*Pa*; (**B**) effects of temperature on the activity of recombinant D2HGDH*Ec* and D2HGDH*Pa*; (**C**) temperature stability of recombinant D2HGDH*Ec* and D2HGDH*Pa*. D2HGDH*Pa* is depicted by dark circles (●). D2HGDH*Ec* is depicted by open circles (○).

**Figure 8 microorganisms-10-01766-f008:**
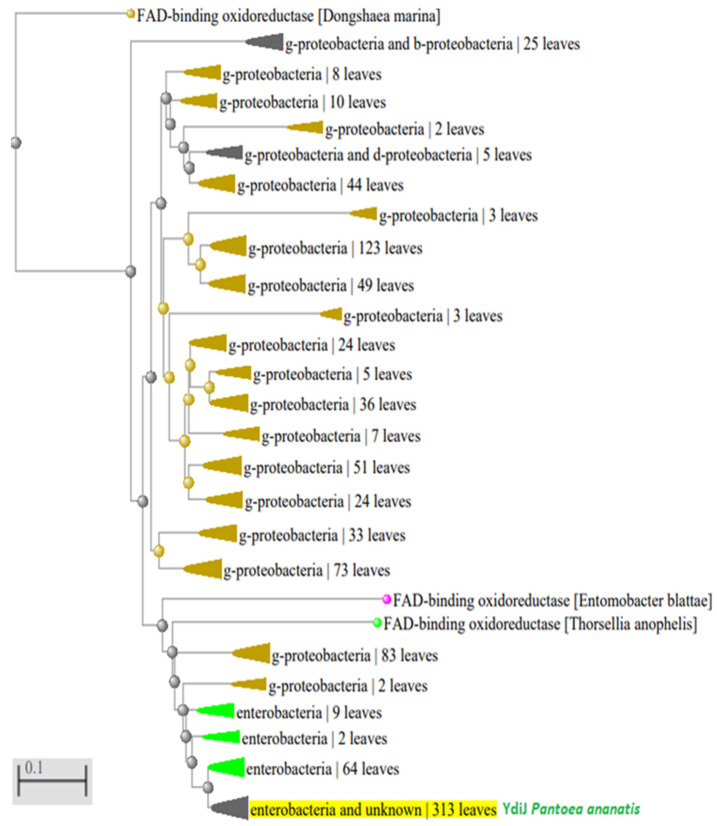
Phylogenetic analysis of D2HGDH*Pa* distribution in *Proteobacteria* phylum. Green color: *Enterobacteriaceae*; black-yellow color: gamma *Proteobacteria*; fuchsia color: alpha *Proteobacteria*.

**Figure 9 microorganisms-10-01766-f009:**
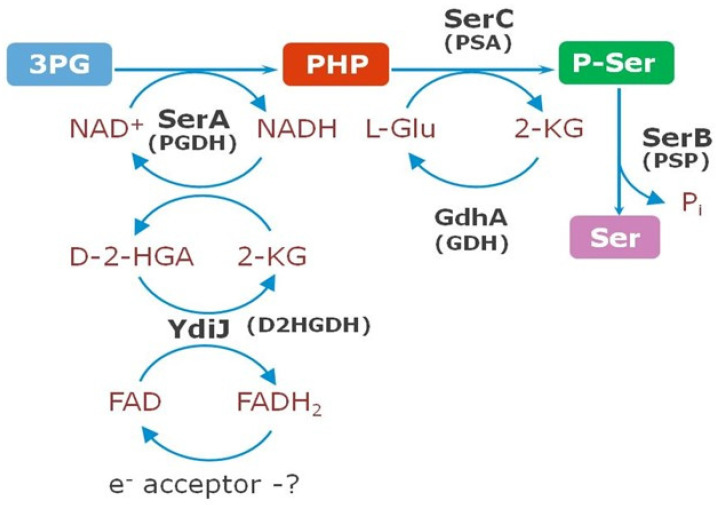
Schematic diagram of L-serine biosynthesis in *P. ananatis* and *E. coli*, associated with recycling of D-2-HGA by YdiJ (D2HGDH). GDH—glutamate dehydrogenase; P_i_—inorganic phosphate.

**Table 1 microorganisms-10-01766-t001:** Bacterial strains and plasmids used in this study.

Strain or Plasmid	Description	Source or Reference
** *Escherichia coli* **		
MG1655	Wild type	VKPM ^a^
DH5α	F^−^ φ80*lac*ZΔM15 Δ(*lac*ZYA-*arg*F)U169 *rec*A1 *end*A1 *hsd*R17(r_K_^–^, m_K_^+^) *pho*A *sup*E44 λ^–^*thi*-1 *gyr*A96 *rel*A1	Novagen
BL21(DE3)	F^–^*omp*T *hsd*S_B_ (r_B_^–^, m_B_^–^) *gal dcm* (DE3)	Novagen
MG1655 P_nlp8_-*ydiJ_Ec_*	MG1655 with enhanced expression of *ydiJ*	This study
MG1655 Δ*ydiJ*::Km	MG1655 with inactivated *ydiJ*	This study
** *Pantoea ananatis* **		
SC 17 (AJ13355)	Wild type	[32]
SC17(0)	Mutant resistant to lambda Red recombinase selected from SC17	[33]
SC17(0) Δ*ydiJ*::Km	SC17(0) with inactivated *ydiJ*	This study
SC17(0) P_nlp8_-*ydiJ_Pa_*	SC17(0) with enhanced expression of *ydiJ*	This study
**Plasmids**		
pMIV-5JS	Vector, SC101 ori, Cm^r^ empty vector	[34]
pMIV-P_nlp8_*serA348(Ec)*	*serA348* (encodes feedback-resistant enzyme [35]) from *E. coli* cloned in pMIV-5JS; the gene is flanked by P _nlp8_ (constitutive derivative of *nlpD* promoter) and T*_rrnB_* (*rrnB* terminator)	This study
pSTV29	Cloning vector; Cm^r^	TaKaRa Bio
pET28b(+)	Expression vector, Km^r^	Novagen
pET-*ydiJ*_Pa1	pET28b(+) with cloned *ydiJ* from *P. ananatis* with 6His on N-end	This study
pET-*ydiJ*_Pa2	pET28b(+) with cloned *ydiJ* from *P. ananatis* with 6His on C-end	This study
pET-*ydiJ*_Ec1	pET28b(+) with cloned *ydiJ* from *E. coli* with 6His on N-end	This study
pET-*ydiJ*_Ec2	pET28b(+) with cloned *ydiJ* from *E. coli* with 6His on C-end	This study
pMW-λattL-KmR-λattR	Source of antibiotic resistance marker Km	[33]
pMW-λattR-KmR-λattL-P_nlp8_	Source of antibiotic resistance marker Km (for enhancing gene expression)	This study
pRSFRedTER	For red-dependent integration of PCR fragments in *Pantoea ananatis*	[33]
pKD46	For red-dependent integration of PCR fragments in *E. coli*	[36]

^a^—VKPM, The Russian National Collection of Industrial Microorganisms (WDCM No. 588); Km^r^—resistance to kanamycin; Cm^r^—resistance to chloramphenicol.

**Table 2 microorganisms-10-01766-t002:** Steady-state kinetic parameters of known D2HGDHs toward D-2-HGA.

Enzyme Family	Mw Calculated	K_m_ (D-2-HGA)	V_max_ (D-2-HGA)	*k_cat_*	*k_cat_* /K_m_	Reference
**VAO/PCMH Flavoprotein** [43,44]	**Da**	**μM**	**μM/min·mg ^−1^**	**s^−1^**	**s^−1^/mM^−1^**	
D2HGDH *P. stutzeri*	51,086.14	170 ± 20	4.56 ± 0.60	7.9 ± 1.05	45.4	[14,20]
D2HGDH *A. thaliana*	51,286.52	~580	NR	~0.8	1.37	[27]
Dld2 *S. cervisiae*	59,282.45	28 ± 8	NR	0.18 ± 0.03	7	[18]
Dld3 *S. cervisiae*	55,241.00	130 ± 9	NR	6.6 ± 0.5	50 ± 2	[18]
D2HGDH *P. aeruginosa*	51,286.52	60	NR	11.2± 0.4	186	[25]
D2HGDH *A. denitrificans*	50,405.42	31.6	40.6	6.9	215	[21]
D2HGDH *R. solanacearum*	50,444.64	433	NR	4.86	11	[26]
D2HGDH *Homo sapiens*	56,416.06	38 ± 0.3 120 ± 10	NR 2.29 ± 0.03	0.51 ± 0.01 1.98 ± 0.03	13 17	[29] [28]
**Fe_4_S_4_ FAD-linked oxidoreductase** [42]						
D2HGDH (YdiJ) *E. coli*	113,247.68	83 ± 5	1.15	13.9	170	**This study**
D2HGDH (YdiJ) *P. ananatis*	113,453.43	208 ± 10	1.17	9.7	50	**This study**

NR—not reported.

**Table 3 microorganisms-10-01766-t003:** Effect of metal ions on the activity of D2HGDH*Ec* and D2HGDH*Pa*.

Metal Ions	Relative Activity (%)	
	*E. coli*	*P. ananatis*
None	100.0	100.0
Mg^2+^	95.0	93.0
Zn^2+^	73.0	34.0
Mn^2+^	0	0
Co^2+^	0	0
Ni^2+^	0	0

## Data Availability

The analyzed data presented in this study are included within this article. Further data are available on reasonable request from the corresponding author.

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
