# Peer review of "Revealing a New Family of D-2-Hydroxyglutarate Dehydrogenases in Escherichia coli and Pantoea ananatis Encoded by ydiJ"

_microorganisms, 2022, doi:10.3390/microorganisms10091766_

Round 1
Reviewer 1 Report
This article describes the search of enzymes able to catalyze de transformation of d-2-hydroxyglutarate (D-2-HGA) into 2-ketoglutarate in bacteria belonging to the Enterobacteriaceae family. The authors determined that Pantoeae ananatis could use D-2-HGA as a sole carbon source. In order to identify the responsible gene allowing the use of D-2-HGA, a multi-copy genomic libraries was created and screened, leading to the identification of a ydiJ gene. A ydiJ gene with high similarity was also described to be present in E. coli, being its translated protein described as putative cytosolic FAD-linked oxidoreductase with unknown function. Inactivation of this gene in both E. coli and P. ananatis led to a significant accumulation of D-2-HGA, hence suggesting its participation in the D-2-HGA metabolism. Thus, both genes were sequenced, cloned, and expressed in Escherichia coli. Subsequently, both obtained proteins were purified, their d-2-hydroxyglutarate dehydrogenase (D2HGDH) activity demonstrated and their biochemical characterization performed.
From my point of view, this article could be published, although I would suggest addressing the following issues:
- A reaction scheme of the pathways described including the chemical structures of the compounds and in which step acts each enzyme would be very helpful for the reader.
- The Michaelis-Menten kinetics curves performed for the calculation of the kinetic parameters of both enzymes should be included, at least in the SI.
- Concerning the effects of metal ions on D2HGDHs activity, taking into account the inhibitory effects that some metal ions have on the proteins and that some metal-dependent proteins get inhibited during purification by Ni2+ metal affinity chromatography, I would suggest to study if Ni2+ affects the activity of the enzymes and if there is any loss of activity during purification due to a potential inhibitory effect of loose Ni2+ ions.
- Many times the blank space between number figures and units is missing.
- The microorganism name’s abbreviation from the restriction enzymes should be written in italics.
- Line 215: French pressure” should be either “French pressure cell press” or “French press”.
- Line 253: pH of the Tris/HCl buffer should be stated.
- Line 263: A reference should be included in the sentence “12% SDS-PAGE was carried out as described elsewhere” to clarify where that information can be found.
Author Response
Dear Reviewer,
Thank you so much for comments and valuable suggestions to our manuscript entitled “Revealing a new family of D-2-hydroxyglutarate dehydrogenases in Escherichia coli and Pantoea ananatis encoded by ydiJ.” Please find our responses to your comments below:
- A reaction scheme of the pathways described including the chemical structures of the compounds and in which step acts each enzyme would be very helpful for the reader.
Thank you. We will add reaction scheme of described pathway for better text understanding in the revised manuscript!
- The Michaelis-Menten kinetics curves performed for the calculation of the kinetic parameters of both enzymes should be included, at least in the SI.
Thank you. We will add Lineweaver–Burk plot for the purified D2HGDHEc and D2HGDHPa toward D-2-HG in SI in the revised manuscript.
- Concerning the effects of metal ions on D2HGDHs activity, taking into account the inhibitory effects that some metal ions have on the proteins and that some metal-dependent proteins get inhibited during purification by Ni2+ metal affinity chromatography, I would suggest to study if Ni2+ affects the activity of the enzymes and if there is any loss of activity during purification due to a potential inhibitory effect of loose Ni2+ ions.
We have checked out effect of Ni2+ ions on D2HGDHs activity, it completely inhibited both of them. We will add this result in the corresponding Table 3 in revised manuscript.
During purification procedure we have observed only about 23% loss of total activity of D2HGDHs which is quite common for another type of enzymes, according to our experience, purified by Ni2+ metal affinity chromatography method. So, we guess there is no problem with loose Ni2+ ions during purification. Final buffer was quickly changed after enzymes purification by Vivaspin 20 concentrator.
- Many times the blank space between number figures and units is missing.
Will fix it in the revised manuscript!
- The microorganism name’s abbreviation from the restriction enzymes should be written in italics.
Will fix it in revised manuscript!
- Line 215: French pressure” should be either “French pressure cell press” or “French press”.
Thank you. Will fix it in the revised manuscript!
- Line 253: pH of the Tris/HCl buffer should be stated.
Will fix it in the revised manuscript!
- Line 263: A reference should be included in the sentence “12% SDS-PAGE was carried out as described elsewhere” to clarify where that information can be found.
As I assume almost 80% scientific articles does not describe “12% SDS-PAGE” in “Matherials and Methods” section and does not have common reference to the original paper of “Laemmli, U.K. Nature 1970, 227, 680–685” because this method well known and standard for protein analysis. It can be found in the web elsewhere.
We can add reference (Complete protocol for SDS-PAGE from Bio-Rad) in the revised manuscript if it really required: https://www.bio-rad.com/webroot/web/pdf/lsr/literature/Bulletin_6201.pdf
Best regards,
Michael Kiriukhin
Reviewer 2 Report
The manuscript “Revealing a new family of D-2-hydroxyglutarate dehydrogenases in Escherichia coli and Pantoeae ananatis encoded by ydiJ.” Is a nice piece of work and should be published after modifications.
The first point is a presentation point. The authors should provide a scheme with the reactions and the structures so that the reader can follow the different conversions more easily. This would also help to get a clearer picture which activity the authors are searching.
In this context a short description of the differences between type I, II and III of the PHGDH’s would be interesting and helpful to have better impression of the novelty of the authors finding described in this manuscript.
It is remarkable that the authors find an entirely new structure for a known conversion, a nice example of convergent evolution.
This leads to a key question. The authors describe that the new enzymes are Fe4S4 containing enzymes. These Fe4S4 clusters are typically not very stable and can be redox sensitive. The authors do not describe that they have to exclude oxygen. Please provide more detail on this. How stable is the isolated enzyme over time and does oxygen play a role?
Author Response
Dear Reviewer,
Thank you so much for comments and valuable suggestions to our manuscript entitled “Revealing a new family of D-2-hydroxyglutarate dehydrogenases in Escherichia coli and Pantoea ananatis encoded by ydiJ.” Please find our responses to your comments below:
The manuscript “Revealing a new family of D-2-hydroxyglutarate dehydrogenases in Escherichia coli and Pantoeae ananatis encoded by ydiJ.” Is a nice piece of work and should be published after modifications.
The first point is a presentation point. The authors should provide a scheme with the reactions and the structures so that the reader can follow the different conversions more easily. This would also help to get a clearer picture which activity the authors are searching.
Thank you. We will add reaction scheme of described pathway for better text understanding in the revised manuscript!
In this context a short description of the differences between type I, II and III of the PHGDH’s would be interesting and helpful to have better impression of the novelty of the authors finding described in this manuscript.
Thank you. We have intentionally gave only short description of Type I, II and III of the PHGDH’s because their properties were thoroughly described and analysed in details by G.A. Grant, “D-3-Phosphoglycerate dehydrogenase, Front. Mol. Biosci. 5: 110 (2018), doi: 10.3389/fmolb.2018.00110”. We have referred to this article. The purpose of our paper is to demonstrate that microorganisms, possessing of Type II PHGDH required system for recuperation of D-2-HGA, produced from 2-ketoglutarate. By example, Corynebacterium glutamicum has Type I PHGDH, which does not produce D-2-HGA from 2-ketoglutarate and does not have neither D2HGDH nor Ydij. I guess complete description of all types of PHGDHs will mislead readers and overloading our paper if we add it.
It is remarkable that the authors find an entirely new structure for a known conversion, a nice example of convergent evolution.
Thank you, we also excited about it!
This leads to a key question. The authors describe that the new enzymes are Fe4S4 containing enzymes. These Fe4S4 clusters are typically not very stable and can be redox sensitive. The authors do not describe that they have to exclude oxygen. Please provide more detail on this. How stable is the isolated enzyme over time and does oxygen play a role?
Thank you for your comment. We knew that Fe4S4 clusters are typically not very stable and can be redox sensitive. When we worked with D2HGDHs, we have only used degassed buffer for purification, nothing special. During purification procedure we have observed only about 23% loss of total activity of D2HGDHs which is quite common for another type of enzymes, according to our experience, purified by Ni2+ metal affinity chromatography method. Both enzymes are quite stable over 6 months (they stored in aliquots at -70C in 50 mM Tris-HCl, pH7.5, and 10% glycerol). We suppose that Fe4S4 of GlpC domain could oxidized and lost its ability to accept electrons, but because for activity measurement artificial electron acceptor is used (DCIP and PES) we could not said is it still active or not.
Best regards,
Michael Kiriukhin